# Effects of γ-Irradiation on Mating Behavior of Red Palm Weevil, *Rhynchophorus ferrugineus* (Olivier, 1790) (Coleoptera: Dryophthoridae)

**DOI:** 10.3390/insects14070661

**Published:** 2023-07-24

**Authors:** Massimo Cristofaro, Chiara Fornari, Flaminia Mariani, Alessia Cemmi, Michèle Guedj, Mohamed Lahbib Ben Jamaa, Meriem Msaad Guerfali, Elisabeth Tabone, Robert Castellana, Raffaele Sasso, Sergio Musmeci

**Affiliations:** 1Biotechnology and Biological Control Agency (BBCA), Via Angelo Signorelli 105, 00123 Rome, Italy; m.cristofaro55@gmail.com (M.C.); fornari.1763472@studenti.uniroma1.it (C.F.); miniguedj@gmail.com (M.G.); 2Department of Environmental Biology, University of Rome “La Sapienza”, 00185 Rome, Italy; 3ENEA, Casaccia Research Center, FSN-FISS-SNI Laboratory, Via Anguillarese 301, 00123 Rome, Italy; alessia.cemmi@enea.it; 4Direction Générale de la Santé Végétale et du Contrôle des Intrants Agricoles (DGSVCIA), 30 Rue Alain Savary, Tunis 1002, Tunisia; benjamaaml@gmail.com; 5Institut National de Recherches en Génie Rural, Eaux et Forêts (INRGREF), Université de Carthage, Rue Hédi EL Karray El Menzah IV, Tunis 1004, Tunisia; 6Laboratory of Biotechnology and Nuclear Technologies, LR2016CNSTN01, Centre National des Sciences et Technologies Nucléaires (CNSTN), Technopole Sidi Thabet, Tunis 2020, Tunisia; msaad_tn@yahoo.fr; 7INRAE UEVT, Laboratoire Biocontrôle, 90 Chemin Raymond, 06160 Antibes, France; elisabeth.tabone@inrae.fr; 8Progetto Phoenix, Centre de Recherche sur le Patrimoine (CRP) 13 rue Victor Hugo, 06110 Le Cannet, France; robert.castellana@laposte.net; 9ENEA, Casaccia Research Center, SSPT-BIOAG-SOQUAS Laboratory, Via Anguillarese 301, 00123 Rome, Italy; raffaele.sasso@enea.it (R.S.); sergio.musmeci@enea.it (S.M.)

**Keywords:** red palm weevil, SIT, behavioral bioassays, insect quality, area-wide

## Abstract

**Simple Summary:**

Red palm weevil (RPW) *Rhynchophorus ferrugineus* (Olivier, 1790) is a highly invasive species of Southeast Asia and Melanesia origin that has spread widely in the Middle East and the Mediterranean area. Its larvae cause extensive damage to several palm species in the Arecaceae family, many of which are economically important for agricultural and ornamental purposes. For this reason, many studies are investigating sustainable and effective management strategies, including the Sterile Insect Technique (SIT). In this study, behavioral bioassays have been carried out in laboratory to evaluate if sterile RPW adult males are able to sexually compete with fertile males in no-choice and choice conditions. Results confirmed that irradiation does not induce any negative effects on the mating behavior and performance of sterile RPW males.

**Abstract:**

Red palm weevil (RPW) *Rhynchophorus ferrugineus* (Olivier 1790) is a highly invasive species originating from Southeast Asia and Melanesia. Over the past 30 years, this alien pest has spread extensively in the Middle East and the Mediterranean basin. Its endophagous larvae feed on various palm species, causing significant damage that leads to the death of palm trees. Controlling RPW infestations is challenging due to their gregarious nature and the lack of detectable early symptoms. Systemic insecticides are effective means of control, but their use in urban areas is prohibited and resistance can develop. Considering alternative options with minimal environmental impact, the Sterile Insect Technique (SIT) has been explored. Previous research has shown that male RPWs irradiated at 80 Gy or higher achieve full sterility. This study aimed to investigate in laboratory conditions whether RPW sterile males (irradiated at 60 and 80 Gy) could compete sexually with non-irradiate males. Laboratory bio-assays under both no-choice and choice conditions assessed sexual performance in terms of number of matings, mating duration and time elapsed until the first mating. The results confirmed that irradiation does not negatively affect the mating performance of sterile males, demonstrating their ability to compete successfully with non-irradiated males in both experimental setups.

## 1. Introduction

Red palm weevil (RPW), *Rhynchophorus ferrugineus* (Olivier, 1790) (Coleoptera, Dryophthoridae), is an invasive insect that feeds on various palm species, in the Arecaceae family. Originally from Southeast Asia [1,2], it has rapidly spread to the Middle East and, subsequently, to the Mediterranean Basin, Australia, China, Japan, California, and more recently to the Caribbean [3]. Similar to many invasive species, in areas of recent colonization populations of RPW have stabilized, lacking effective natural enemies [4]. In Italy, for example, this insect was first reported in 2005 [5], after being introduced through illegal palm trade, which were later found to be infested with larvae of the beetle. To date, it is possible to find this beetle in most regions of Italy [6].

The presence of RPW in Italy has caused numerous problems, including economic losses, environmental damage, and the use of chemical pesticides for pest control. In addition, it poses a threat to endemic Mediterranean species such as the Mediterranean fan palm (*Chamaerops humilis* L.) and the Canary Island date palm (*Phoenix canariensis* Chabaud). The disappearance of these palms is a concern, and there is a risk of RPW permanently infesting the date palm (*Phoenix dactylifera* L.), economically important in North African and Middle Eastern countries bordering the Mediterranean [7]. Both palm species, *P. canariensis* and *P. dactylifera*, are phylogenetically close, in fact, studies have already shown how the RPW is able to oviposit, develop and induce the death of date palm [7,8].

The control of RPW is not an easy task, as the biological cycle of this species does not allow early monitoring of the damage caused by the endophagous larvae that live and feed exclusively inside the palm. By the time first symptoms of the attack are visible, such as the loss of the vegetative apex and the umbrella-like bearing of the leaves, the infestation may be well-established making it harder to control [9]. Furthermore, the reproductive success of the RPW also allows this species to spread rapidly: females can lay several hundred eggs during a long oviposition period [10], managing to produce fertile eggs up to one month after copulation [11]. These behavioral and biological characteristics make RPW difficult to manage with conventional methods such as the use of insecticides [12]. Although effective pesticides have been identified [13], it is not, however, easy to reach the target insects within the plant with pesticides that, in any case, may have harmful effects on human health and environment. 

More appropriate strategies based on territorial approach, such as the Sterile Insect Technique (SIT), could improve the likelihood of successful RPW control, especially considering physically and ecologically isolated or susceptible areas [14], alone or in combination with other biological control strategies [15]. This technique is an area wide form of pest management which is based on massive releases of adults sterilized by ionizing radiation, within a population of the same pest species whose ability to mate is as minimally compromised as possible, in order to introduce a genetic load that could lead to its decrease or even eradication [16,17,18].

In the 1950s, the successful application of the SIT led to the eradication of the new-world screwworm fly, *Cochliomyia hominivorax*, from the Americas [19]. Since then, the SIT has been employed for managing various insect species [20,21].

One primary challenge of using SIT as an insect control strategy involves the development of a laboratory strain that is both reproductively compatible and competitive with the target population. The processes involved in strain development for SIT include the optimization of a sterilization or irradiation method [16], which could change the corresponding genotypes and phenotypes of an insect strain [22], resulting in reduced mating competitiveness with the target population or sometimes even incompatibility.

Although several laboratory [23,24,25,26,27,28] and field studies [23,29] have already highlighted the applicability of the SIT to control RPW, some aspects of the reproductive sphere of this species, such as polyandry, post-copulative sperm selection mechanisms and the gregarious behavior [30] might make the use of this technique unsuitable for its control. In fact, according to Lance and McInnis [25], the success of infertile mating between sterile males and wild females may be lost if the females later also re-pair with wild males and select the latter’s sperm for fertilization.

However, even in polyandric species, the SIT remains applicable [30]. The successful application of SIT requires that sterile males can compete and mate successfully with their wild counterparts [25], maintaining mating propensity, ability to locate a mate, copulate and inseminate, despite the physiological damage usually caused by radiation.

In the specific case of *R. ferrugineus*, previous studies demonstrated that γ-irradiation has important effects on the mortality of the males: the longevity of males irradiated at 60 and 80 Gy was reduced to 2–3 weeks compared to more than 120 days in the control [12]. In addition, irradiated males at 80 Gy were able to induce full sterility of non-irradiated virgin females with whom they mated [12]. More recent studies, using wild red palm weevils collected in the field have shown that the eggs laid by females turn out to be exclusively those fertilized by the last male encountered, according to the post-copulatory selection mechanism called last-male sperm precedence [31,32,33]. Same results have been achieved using wild-type red palm weevils: wild field collected *R. ferrugineus* irradiated males (80 Gy) were able to induce the full sterility in wild non-treated and already mated females, captured using Rhyncho-Traps^®^ triggered with an aggregation pheromone and ethyl acetate [31]. However, in a SIT application context, the most suitable dose of irradiation should be considered a trade-off between effective sterility and male competitiveness [34]: in fact, as the dose increases, sterility increases but so do the consequences on male quality, longevity and mating competitiveness [12,31,35]. It is therefore necessary for dose selection to possess information on the influence of dose on sterility and indicators of insect quality.

Thus, the following study aims to further behavioral laboratory bioassays in order to evaluate the competitiveness of irradiated males in mating in comparison with fertile males, both in no-choice and choice conditions. 

To exclude the possible side-effects of the selection of genotypes better adapted to captivity, we decided to use only wild-type RPW adults, keeping the γ-irradiation as the only variable that would interfere with their mating competitiveness.

Considering that *R. ferrugineus* is a gregarious insect, and therefore male–male interactions are to be expected [6,36], the objective of this study was to evaluate in a promiscuous context (a choice-test, confining one female with two males, one fertile and the other sterile), which one will mate first and whether the mating preference will remain stable during the 12 days of the experiment. 

In addition, we focused our efforts to detect eventual differences in mating behavior between fertile and irradiated RPW individuals, and whether these differences might induce a mechanism of preference by females.

## 2. Materials and Methods

### 2.1. Insects and Male Irradiation

Following a protocol used in previous studies [6,31], wild adults of *R. ferrugineus* were collected in Sicily, near Palermo and in Pantelleria Island, between November 2020 and August 2022, through the use of Intrachem Rhyncho Trap^®^ triggered with the aggregation pheromone Ferrugineol and ethyl acetate and placed 400–500 m from healthy *Phoenix canariensis* or *P. dactylifera* palms or in the vicinity of *Phoenix* spp. palms already attacked by *R. ferrugineus*. Since previous behavioral studies in gregarious conditions clearly showed that mating patterns in RPW males and females are independent from the physiological status, age and size of the individuals, we used for the tests almost all the adults collected (discarding only the weevils that showed signs of damage or deformity) [6,36].

Laboratory rearing consisted in field collected adult individuals placed inside transparent plexiglass terrariums measuring 35 × 20 × 30 cm and fed with a diet consisting of apple slices (Golden Delicious variety) mixed with coconut fiber. Males and females were kept in separate containers. No more than 15 individuals were placed in each container to ensure that they did not experience stress from overcrowding.

Apple slices were replaced three times a week to prevent mold growth and allow weevils to always have a fresh food source. 

Irradiation was carried out at the ENEA “Calliope” facility by exposing wild adult males to a source of Cobalt-60 γ rays [37] at a rate of 12.5 Gy min^−1^. For this study, irradiation doses at 60 Gy and 80 Gy were selected as the most optimal according to a previous screening on the longevity-sterility ratio [12]. 

### 2.2. Experimental Design

Two types of behavioral bioassays were carried out, which involved confining adult individuals in 500 mL transparent glass jars, 12 cm in diameter, 5 cm of height, covered with a 680 μm white polyester mesh and with sand at the bottom:No-choice: wild female + irradiated male (60 or 80 Gy). A control was set up with a wild female and a wild male.Choice: wild female + irradiated male (60 or 80 Gy) + wild male. A control was set up with one wild female and two wild males.

For each combination, 15 replicates were performed. In choice assays involving the co-presence of two males, markings with 2 different colors of non-toxic natural water-based paint (Benecos brand) were performed to distinguish the male irradiated from the fertile male. Also in the choice assay control set, the two fertile males were marked with two different colors. To avoid side effects due to an eventual interference of one of the two marking colors, in 50% of the replicates the marking colors were switched. In order to record reproductive parameters and their possible changes in relation to the time elapsed since irradiation, observations of adults’ behavior occurred at regular intervals (3, 6, 9 and 12 days) after irradiation of the males, to compare the eventual differences in terms of competitiveness between fertile and irradiated males over the time. The 12-day-timespan was selected following the previous screening on the effects of gamma-irradiation on the physiology of red palm weevil, showing that the longevity of weevils irradiated at 60 and 80 Gy was reduced up to just 2–3 weeks [12]. 

For the start of the experiment, first all males were added to each of the test containers and then all females, to ensure that pair formation was as rapid and simultaneous as possible for all replicates. Visual observations were made every minute for a duration of 4 h, always at the same time of the day, from 11:00 a.m. to 03:00 p.m., considered the hours of highest sexual activities in *R. ferrugineus* [36].

Because pre-mating interactions do not necessarily lead to successful mating, the occurrence of insemination was assessed by visual observation of the aedeagus in the female genital opening for a time longer than 30 s [36].

The following behavioral data were recorded: The interval from T_0_ (time when the individuals were placed in the containers) and the first mating, to compare the mating competition between fertile and sterile males;Duration of each mating;The total number of mating events that occurred in 4 h.

At the end of each observation, females were isolated in glass jars (6 cm in diameter) labeled to trace them back to the mating jar from which they were taken. A quarter apple wedge each was provided as a food source. Females were kept in these jars until the next observation, when they were added back to the same container from which they were taken. 

### 2.3. Data Analysis

#### 2.3.1. No-Choice Test

A glmer analysis (generalized mixed model with random effects; package lme4, [38] in the statistical environment R [39]) was performed on the response variable mating frequency that was recorded as event occurrence having a binomial distribution 0, 1. The mating duration, the number of mating events per day and the time elapsed until the first mating were also considered as response variables, given the importance of these behavioral parameters for the reproductive fitness evaluation in the preparatory phase of SIT studies. The response variables mating duration and time elapsed before the 1st mating, were expressed as number of minutes, while the number of mating events per day was expressed as a count. The model design was chosen based on the optimal parsimony principle (AIC and BIC estimators) and on significance of the overall effects. Three fixed effects (the explanatory variables) were considered as factors: the dose applied to the irradiated male (0, 60 and 80 Gy), the experimental phase in hours (from 1 to 4), and the day of the experiment (at 3, 6, 9 and 12 days from male irradiation), to analyze any variation in the behavioral and physiological state of insects during the experiment. The couple used in the experiment was considered as the random effect, and the factor ‘hours’ was considered nested into the random effect.

#### 2.3.2. Choice Test

Also in this case, a glmer model was applied using the same explanatory variables as fixed effects (dose, day and hour) on the same response variables as in the case of no choice test. The effect of the males 1 and 2 was considered nested into the dose treatment. In the case of dose zero, males were both not irradiated and were used as a separate control. The experimental unit with the three individuals to be tested was considered as a random effect. 

Two main differences were made in comparison with the no-choice test: (1) For the response variable mating frequency, the effects of the days and of the hours were analyzed separately in order to simplify the analysis and to avoid convergence problems in the model. Mating frequency was expressed here as number of minutes per unit of time (days or hours) and a negative binomial distribution was applied. (2) In the case of the response variable ‘first mating’, two models were considered: In the first model a binomial distribution was applied within the glmer model. Thus, the response variable consisted if the first mating occurred with an irradiated male or not. In this model was considered also the case where one of the two males did not mate at all during the 4 h of the experiment. The second model was performed in order to analyze the case when both irradiated and healthy males were competitive enough to mate once at least. Thus, in the second model, only the cases where both males mated once at least were analyzed. The number of minutes elapsed before the 1st mating were used as response variable and a negative binomial distribution was applied into the glmer model. 

The Tukey’s test for mean separation was performed in the statistical environment R by the multcomp package [40]. In the text the means are always followed by the standard errors. 

## 3. Results

### 3.1. No-Choice Test

#### 3.1.1. Mating Occurrence

The mean values of mating proportion (expressed as event occurrence 0, 1) in relation to the Gy dose applied and to the days of experiment, are reported in Figure 1 and in Table A1. As Table 1 shows, mating did not occur for the majority of the time.

A significant effect of the irradiation dose was observed in the ANOVA estimates performed on the overall effects of the glmer model (χ^2^ = 8.15, df = 2, *p*-value = 0.0170). Even stronger effects were found in the case of the day of experiment (χ^2^ = 15.20, df = 3, *p*-value = 0.0016) and for the hours elapsed from the beginning of the daily experiment (df = 3, χ^2^ = 53.64, *p*-value = 1.3^−11^). As regards the interactions effects, no significant interactions between hours and dose were found, while the interaction dose*day reached statistical significance (χ^2^ = 14.32, df = 6, *p*-value = 0.0263) and for this reason the latter was introduced in the glmer model. 

As Table 2 shows, at 60 Gy dose, insects mated for a higher number of minutes, but these values were borderline significant when compared to the untreated control. However, a significantly higher frequency of mating was recorded at 60 Gy, when compared to the 80 Gy dose (coef = −0.763 z-value = −2.787 *p*-value = 0.0146). 

An inverse trend was found for days and hours factors (Table A1 and Table 2): more mating events occurred in the days following the third day of the experiment, while a general trend towards fewer mating events was found with the passing of hours (Table A1 and Figure 2a).

Surprisingly, the increase in mating events as the days passed was even more evident for the couples with the irradiated males in comparison with the untreated control, especially in the case of 60 Gy dose (Table 1, Figure 2b). For untreated control, a higher increase in mating frequency between the 3rd and the 6th day of the experiment was found in comparison to the 60 Gy dose (Figure 2b; interaction effect: coef = −0.381 z-value = −2.14, *p*-value = 0.0320), but in the following days, mating events were significantly more frequent at 60 Gy compared to the untreated control, especially in the 12th day of the experiment, as verified by the Tukey’s test. In fact, the untreated control reached a maximum in the 6th day of the experiment and then slightly decreased on the next days (Figure 2b; Table A1), while the irradiated ones increased the mating frequencies up to the 9th day (80 Gy) or up to the 12th day (60 Gy). It is worth noting that on the 12th day of the experiment the dose of 80 Gy reached values of mating frequency very similar to the untreated control (Figure 2b; Table 1 and Table A1).

#### 3.1.2. Mating Event Duration

Mean values of the mating event duration in relation to the Gy dose effect and to the days of experiment are reported in Table A1, while the mating events divided in categories of duration are reported in Table A2. Only a borderline significant effect of the dose was observed in the ANOVA model (χ^2^ = 4.87, df = 2, *p*-value = 0.087), while no significant effects of the day (χ^2^ = 4.22, df = 3, *p*-value = 0.238) were found. However, as Table 3 shows, longer mating events were recorded at the 60 Gy dose in comparison to the untreated control (coef = 1.988, z-value = 0.114, *p*-value = 0.0468), while only a borderline significance was observed at the 80 Gy dose (coef = 0.121, z-value = 1.862, *p*-value = 0.0626).

In any case, the differences were small (Table A1; Figure 3a). In particular, mating events longer than 2 min took 29.3% of the mating time at 60 Gy and 27.9% at 80 Gy in comparison to the 20.1% recorded at dose zero.

#### 3.1.3. Number of Mating Events per Day

The mean values of the number of mating events per day in relation to the dose Gy and to the days elapsed are reported in Table A1. A significant effect of the dose Gy was found for this reproductive parameter, according to the ANOVA model estimates (χ^2^ = 7.35, df = 2, *p*-value = 0.0254). A significant effect was found also for the day (χ^2^ = 12.45, df = 3, *p*-value = 0.0060), with a general increase in mating frequency as the days passed (Figure 3b) but looking at the glmer model estimates (Table 4), the dose effects were not significantly different from the untreated control. However, the 60 Gy dose had a number of mating events significantly higher than the dose Gy 80, as verified by the Tukey’s test for means separation. 

#### 3.1.4. Time Elapsed before the First Mating

As regards the results about the time elapsed before the first mating (Figure 4, Table A1 for means and Table A3 for the categories of elapsed time), a significant effect of the dose was found (χ^2^ = 12.31, df = 2, *p* = 0.0021), while only a borderline significance was observed on the day effect (χ^2^ = 7.33, df = 3, *p* = 0.0621) and on the interaction between day and dose (χ^2^ = 12.53, df = 6, *p* = 0.0511).

A significantly shorter time elapsed was observed at the 60 Gy dose in comparison with the dose of 80 Gy when the latter was used as reference level (coef = −1.8689, z-value = 3.504, *p* = 0.0005), but not differences were found for both the tested doses when compared with the dose zero (Table 5). 

A reduction in the time elapsed was observed at the dose of 80 Gy over time, and a statistically significant difference was found between the 3rd day and the 12th day at this irradiation dose (Figure 4b, Table A1). In addition, in the third day of the experiment, the time elapsed before mating was significantly higher at 80 Gy in comparison to 60 Gy (Figure 4b), as verified by the Tukey’s test. It is worth to note that at the 12th day of the experiment at 80 Gy, the time elapsed before mating was very similar to that recorded on the untreated control (Figure 4b), suggesting a better performance of the males irradiated at 80 Gy during the last days of the experiment in comparison to the early phases of the experiment. 

### 3.2. Choice Test

#### 3.2.1. Total Amount of Time Spent in Mating

Insects were sexually inactive most of time during the experiments, as already observed in the no-choice mating experiments (Table 6), with mating occurring only for 7.5% of the experiment duration. 

The total time of mating across the entire duration of the experiment was longer in the irradiated males than the in the fertile ones, both at 60 Gy and 80 Gy, with 47.3 vs. 24.5 min of mating time per couple at 60 Gy and 56.6 vs. 29.8 at 80 Gy. As expected, similar mating frequencies were observed on the two untreated fertile males (28.5 vs. 29.2 min). These outcomes were analyzed by a generalized mixed model with random effects. A statistically significant effect of the treatment within the tested doses was observed, according to the ANOVA model performed on the overall effects of the glmer model (χ^2^ = 14.62, df = 3, *p*-value = 0.00217). Moreover, a significant effect of the day was recorded (χ^2^ = 20.57, df = 3, *p*-value = 0.00013).

Looking at the glmer model estimates calculated for the day and the dose Gy effects (Table 7), a significantly higher number of minutes spent in copulation was observed in the sixth and ninth day of experiment in comparison to the third day, regardless of the tested doses (coef. = 0.701, z-value = 3.58, *p*-value = 0.0003 in the sixth day and coef = 0.577, z-value = 2.92, *p*-value = 0.0035 in the 9th day) (Table A4). 

The irradiated males mated for longer time than untreated ones (coef = 0.760, z-value = 2.818, *p*-value = 0.00484 at 60 Gy; coef = 0.8177, z-value = 3.054, *p*-value = 0.00226 at 80 Gy) (Table 8 and Table A4). Instead, no significant effects were observed on the control (coef = 0.189, z-value = 0.700, *p*-value = 0.484). Irradiated males reached a peak of sexual activity in the 6th day of the experiment (Table 8 and Table A4, Figure 5). 

Regarding the effect of the hours elapsed, a strong effect was observed, according to the ANOVA model (χ^2^ = 35.85, df = 3, *p*-value = 8.05 × 10^−8^). Looking at the model estimates (Table 9), the time spent mating decreased over time, regardless of the applied dose treatment (Table 8 and Figure 6). 

The irradiation treatment had a positive effect on the competitiveness of males in terms of time spent in copulation in comparison to the fertile males regardless the hours elapsed of experiment (at 60 Gy: coef = 0.712, z-value = 2.667, *p*-value = 0.00765; at 80 Gy: coef = 0.753, z-value = 2.832, *p*-value = 0.00462), whereas no significant differences were recorded when both the fertile males competed for mating (Table 9 and Table A5, Figure 6). In particular, during the first hour, the minutes spent in copulation were more than double, at 80 Gy, than with the fertile males. In addition, the gap between irradiated and fertile males, persisted until the end of the experiment both at 80 and 60 Gy (Table 8, and Figure 6), although at lower frequencies with the passing of the hours. Regarding the comparison among the choice experiments, no significant differences were found between the 80 and 60 dose Gy for the minutes spent in mating, as verified by the Tukey’s test. 

#### 3.2.2. Mating Event Duration

No significant differences between the irradiated males and the fertile males for the mating event duration were observed (Table 10), although a borderline significance towards longer mating episodes was found in the case of the irradiated males at 60 Gy in comparison to the fertile males (coef = 0.111, z-value = 1.919, *p*-value = 0.055). Finally, no significant effects of the day, or its interaction with treatment were detected (Table 10 and Table 11, Figure 7a).

#### 3.2.3. Number of Mating Events Per Day

This reproductive parameter was similar to the frequency data (Table 11, Figure 7b), since no relevant differences in mating duration were observed. In fact, looking at the model estimates reported in Table 12, also for this parameter a strong and significant higher number of mating events per day was found for the mating with the irradiated males in comparison to the fertile ones (Table 11), for both 60 and 80 Gy treatments (at 60 Gy: coef = 0.4835, z-value = 3.497, *p*-value = 0.00047; at 80 Gy: coef = 0.572, z-value = 4.252, *p*-value = 2.1 × 10^−5^). At 60 Gy, the number of events was 3.48 ± 0.42 versus 1.97 ± 0.30 events observed with untreated males, while at 80 Gy, irradiated males mated 4.50 ± 0.52 times versus the 2.53 ±0.47 times recorded for the fertile males. 

#### 3.2.4. Occurrence of First Mating and Time Elapsed before the First Mating

As regards the type of first mating occurrence (irradiated or fertile male), no significant effects were found, although the 60 Gy treatment showed a borderline significance (Table 13) toward more events of first mating with the irradiated male than with the untreated male (Figure 8), recording 25 events compared to 12 observed with the fertile male (coef = 0.974, z-value = 1.740, *p*-value = 0.0819). 

The male irradiated at 80 Gy mated more frequently before the fertile male (23 cases vs. the 15 cases of the untreated male), but statistical significance was not reached, probably also due to the small sample size. Conversely, regarding the time elapsed before the first mating (Table 11; Figure 8), a much shorter time was recorded on the irradiated male at 80 Gy in comparison to the fertile male (Table 14), (coef = −1.580, z-value = −3.631, *p*-value = 0.00028).

## 4. Discussion

Sterile insect technique is based on the propensity of fertile females to mate with sterile males which greatly outnumber fertile males [16,17]. Thus, sterile males and their sperm must be competitive, and therefore functional in mating propensity and reproductively compatible [20,32,34]. 

Quality assurance is achieved by conducting behavioral bioassays that assess various parameters, reflecting the insect capacity to survive, interact with its surroundings, and successfully locate, mate with, and fertilize females of the target population [34]. In the past, poor performance of sterile males in terms of mating competitiveness has been always attributed to side effects of irradiation [41,42]; on the contrary, the mass-rearing process can promote genetic drifts, inducing genotypic differences between wild and laboratory populations [43]. Previous studies confirmed that the wild-type irradiated adults, collected in the field by mass trapping, did not differ in terms of fitness and behavior from newly emerged *R. ferrugineus* adults [6,31]. In the present study, all the weevils were wild-harvested, and thus their mating competitiveness would mainly have been influenced by irradiation. It is worth to note that in previous studies RPW adult males collected from field, showed a complete sterility after irradiation, and they were able to induce full sterility even when mated with wild-type already fertilized females [31].

The main objective of the behavioral bioassays carried out for this study was to highlight any differences between the performance of fertile males and sterile males irradiated at two different doses. The selected doses (60 and 80 Gy) were chosen because in previous studies they provided the best response in terms of male lifespan and sterility [12,31]. These behavioral assays not only provided a deeper understanding of the inherent structure of the mating system of *R. ferrugineus*, but also revealed interesting patterns attributable to the effects of irradiation dose, during a 12-day period (day 3, day 6, day 9 and day 12), which correspond to previous data on the longevity of weevils irradiated at 60 and 80 Gy [12].

Excluding the variable “quality of the insects reared in captivity”, statistical analyses of our data clearly confirmed that neither of the two radiation doses tested prevented the irradiated RPW males from mating. This finding excludes the possibility that radiation negatively interferes with the male insect’s sexual performance, preventing the mating or inducing female repulsion. In this work the time spent in mating events was lower compared to time spent in inactivity (Table 1 and Table 6), showing a trait of the RPW female to mate more times for short periods and with more males, according to the polyandrous behavior [6,36]. 

In no-choice conditions, the amount of time spent in mating during the full experiment period was not significantly different between the control males and those at the two irradiated doses. Also for the other variables (number of matings, duration of the first mating and the time elapsed in minutes before the first mating), it was not recorded any particular difference between the control (fertile males) and the irradiated ones (Table 1, Table 2, Table 3, Table 4 and Table 5 and Figure 1, Figure 2, Figure 3 and Figure 4), except for the mating frequency, where the male irradiated at 60 Gy were performing slightly better than the fertile males (Figure 2).

However, notable differences were observed between fertile and sterile males in choice conditions. Both groups of irradiated males exhibited optimal competitiveness against wild males, while females always exhibited a passive behavior during the mating phase, displaying no active preference for a specific male, according to previous data on aggregation behavioral observations [6,36]. 

The differences between the irradiated and fertile males involved several aspects of mating. Except for the duration of the mating, which does not show any significant differences between the irradiated and the fertile male performance, the results for the other two behavioral variables (number of matings and the time elapsed from the beginning of the experiment to the first mating) are showing an evident better performance of irradiated males (Table 6, Table 7, Table 8, Table 9, Table 10, Table 11, Table 12 and Table 13 and Figure 5, Figure 6, Figure 7 and Figure 8). In particular for the number of mating events (Figure 6, Figure 7b and Figure 8a) and for the time elapsed from the beginning of the experiment to the first mating (Figure 8b), the results highlight that irradiated male (in particular the ones irradiated at 80 Gy) were sexually more competitive than fertile ones. This is particularly true, considering previous behavioral studies on RPW carried out in gregarious conditions, where mating context has been recorded as highly promiscuous, with several interactions among all the individuals [6]. Matings were representing the most frequent interactions (between 80.6 and 89.1 of the total interactions), with *R. ferrugineus* males performing frenetic searches for matings in a promiscuous aggregation context [6,40]. 

The aspect correlated to the short time elapsed recorded in choice tests with males irradiated at 80 Gy, needs to be better analyzed: comparing the results in no-choice conditions (no significant differences among the treatments, Figure 4a) with the results in choice conditions (just 5 min to start the mating, Figure 8b), the performance of the males irradiated at 80 Gy when another male is present is showing extremely competitive patterns; this response showing that irradiated RPW males (80 Gy) are clearly more sexually competitive than the fertile ones, confirming that the reported decline in “insect quality” can be related to the mass-rearing, handling and release practices [43,44,45]. Despite to the fact that in Figure 1 (no-choice) and in Figure 6 (choice) the mating frequency in irradiated males is following a negative physiological trend over time, the irradiated males are always performing better than the control (in particular, for the males irradiated at 80 Gy). These results, combined with the data on the time elapsed from the beginning of the experiment and the first mating, clearly showed that: (i) the mating performance of irradiated males does not decline with the approaching of the end of their life; (ii) 80 Gy irradiated males are the most suitable in terms of mating performance, confirming previous data on the effects of the irradiation on the physiology of males of this pest species [12,31].

Even most of SIT programs have been applied to target pests belonging from the Orders Diptera and Lepidoptera [25,41], there are some recent studies have shown the feasibility of including the SIT in area-wide control programs against invasive alien weevils [14], with the possibility even to eradicate the target pest from the territory [46]. 

Particularly for RPW, this pest species exhibits two peculiar physiological and biological characteristics associated with mating that support the use of SIT for its control: (i) the complete absence in the female of a refractory period after mating, along with (ii) last mating male sperm precedence. These two post-mating responses are increasing the probability of fertile females mating with sterile males and decreasing the overall reproductive success of the population [6,31,36]. 

Concern about the suitability of implementing mass rearing facilities for this gregarious target species, new ideas and approaches are under consideration based on mass trapping, irradiation and release of large numbers of sterile males instead of multiplying them in a laboratory [43,45,47]. The classic SIT approach, based on the assessment of large mass rearing facilities is not an easy task for this species, for the long life-cycle (several months), for the presence of cannibalism behavior at the larval stages and for the complicate issue to spin a cocoon as pupation site: our preliminary data show the long life-cycle duration and the cost for the artificial diet as the most crucial aspects [12]. More recent studies show that a semi-artificial diet can be used to rear small-scale laboratory colonies [48], but probably they are not suitable for large scale SIT implementation programs. The alternative that we are taking into consideration when dealing with arthropod pests showing a clear gregarious behavior is a new approach, based on combining mass-trapping and SIT for small scale programs [43,45]. Field traps trigged with the aggregation pheromone can provide large numbers of alive RPW of both genders: keeping (or eliminating) the females and release wild males after irradiation in the environment can be a suitable and sustainable approach in peculiar, well isolated territories, such as the small island of Pantelleria or a date palm oasis in the Sahara Desert. In particular, Sterile Insect Technique can be utilized in date palm growing areas where the distribution of palm trees is regular and continuous but isolated (palm groves in oases). These conditions are ideal for an area-wide approach as pointed out by Klassen [14]. Therefore, it is important to consider a multitrophic scenario that includes agronomic, socio-economic, and biological factors, along with the physiological reaction of the target pest to irradiation.

## 5. Conclusions

The success of an area-wide pest management program that has the SIT as its strategic core relies on setting up a system to release an adequate number of sterile males able to compete with wild males for mating opportunities. Therefore, this work has focused on using sterilizing radiation doses while maintaining the mating competitiveness of sterile RPW males with respect to non-irradiated ones. 

The data presented in this study suggest that irradiated wild-type male adults of *R. ferrugineus*, especially those irradiated at 80 Gy, exhibit strong mating competition behavior when confined in a cage with fertile males. Comparison between the 60 Gy dose, the 80 Gy dose and the control showed several interesting aspects: the doses used did not affect mode and timing of mating: after the first copulation, adults mated again and repeatedly, even more than untreated adults. 

Laboratory tests on competitiveness are generally not entirely reliable in predicting performance and success in the field; this is why more accurate results can be obtained in studies carried out directly in the field. Additional laboratory and confined field tests are also necessary to better understand the interactions in large density weevil conditions. 

## Figures and Tables

**Figure 1 insects-14-00661-f001:**
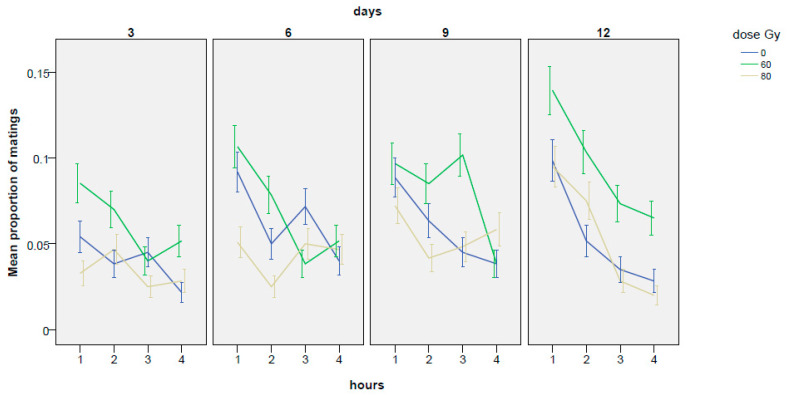
Trends in mating proportion during the 4 h of experiment recorded at the 3rd, 6th, 9th and 12th day; 60 and 80 Gy are compared with the untreated control. The experimental phases are subdivided into periods of 1 h to point out eventual changes in the mating frequency during the experiment. Means and ± standard errors are reported.

**Figure 2 insects-14-00661-f002:**
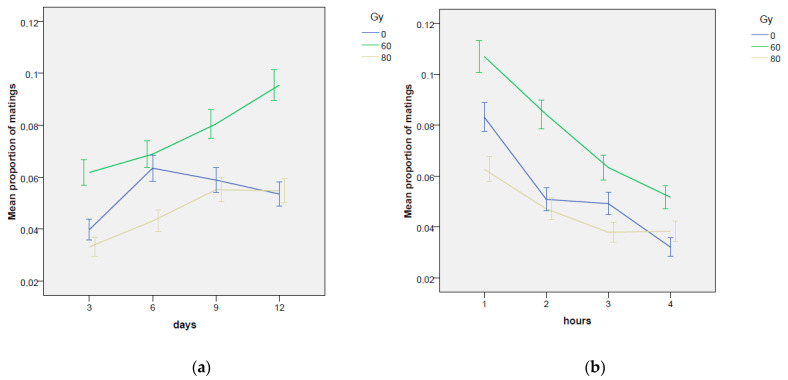
No-choice test. Trends in mating frequency: the effects of the experimental phase (hours) (**a**) and of the day (**b**) are considered separately for the 0, 60 and 80 Gy doses. Means and standard errors are reported.

**Figure 3 insects-14-00661-f003:**
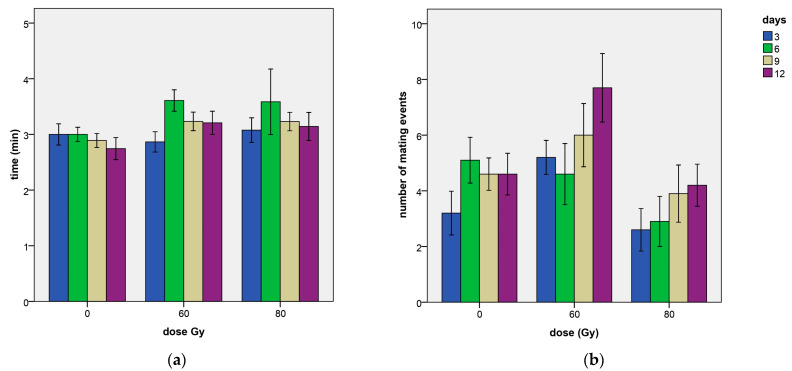
Duration of mating (**a**) and number of mating events per day (**b**) through the days of the experiment in relation to the applied doses of 0, 60 and 80 Gy. Means and standard errors are reported.

**Figure 4 insects-14-00661-f004:**
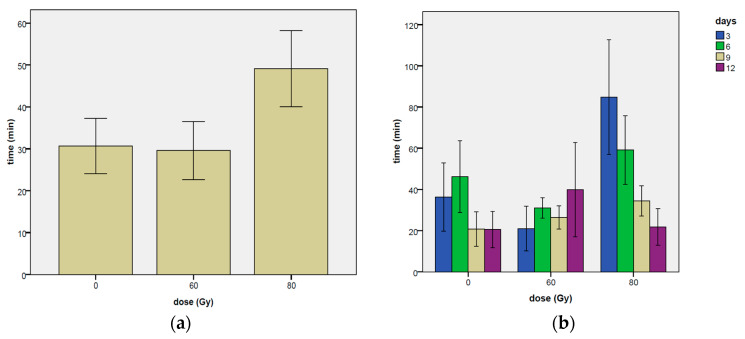
Time elapsed before the first mating per couple observed on the irradiation doses of 0, 60 and 80 Gy. In (**a**) the total means of irradiation doses are shown. In (**b**), data are grouped according to the days of experiment. Means and ± standard errors are reported.

**Figure 5 insects-14-00661-f005:**
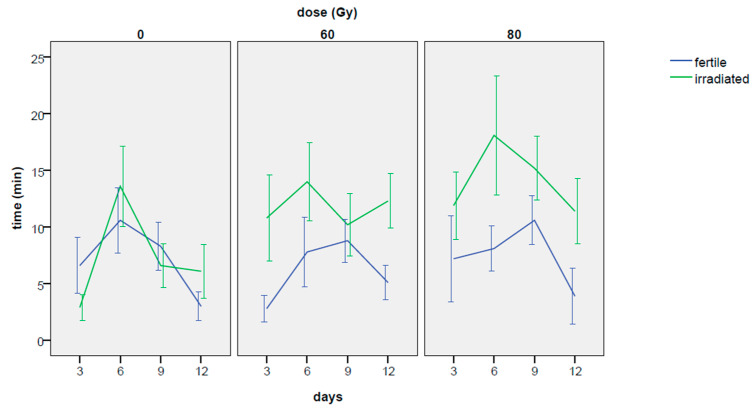
Time spent in copulation during the 12 days of the experiment. The applied doses of 60 and 80 Gy are compared with the untreated control. Both the males at the dose zero are untreated. Means and ± standard errors are reported.

**Figure 6 insects-14-00661-f006:**
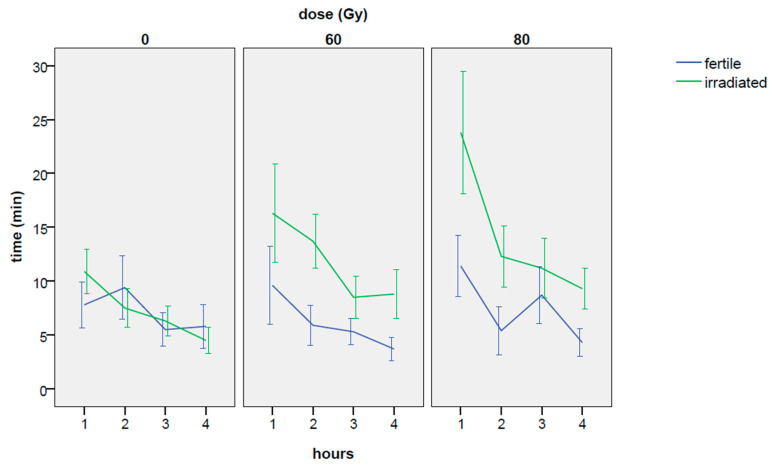
Time spent in copulation during the 4 h of the experiment; means are computed across the 4 days of the experiment. The applied doses of 60 and 80 Gy are compared with the untreated male. Both the males at the dose zero were untreated. Means and ± standard errors are reported.

**Figure 7 insects-14-00661-f007:**
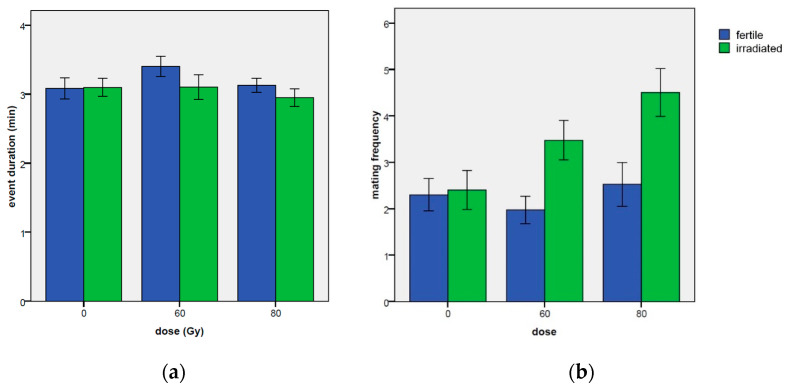
Duration of mating (**a**) and number of mating events per day (**b**) through the days of the experiment in relation to the applied doses of 0, 60 and 80 Gy. Means and ± standard errors are reported.

**Figure 8 insects-14-00661-f008:**
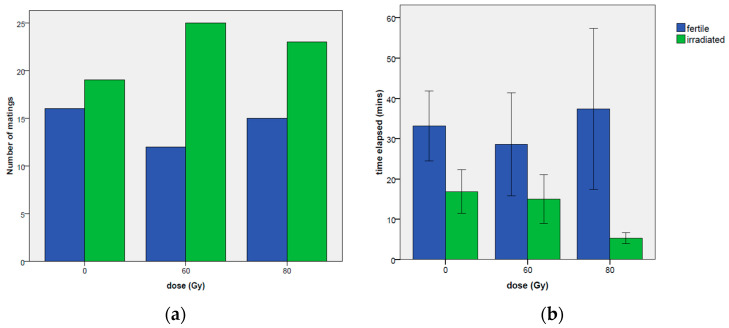
Number of events of first mating at the tested doses for fertile and irradiated males (**a**); time elapsed in minutes before the first mating observed on the irradiation doses of 0, 60 and 80 Gy (**b**) (only the cases when both males mate once at least are considered). Both the males at the dose zero are untreated. Means and ± standard errors are reported.

**Table 1 insects-14-00661-t001:** Total amount of minutes spent in mating or in inactivity in relation to the applied dose of irradiation (0, 60 and 80 Gy). The percentages of amount of time are also reported.

Time	Mating	0 Gy		60 Gy		80 Gy		Total	
Days	Yes/No	Min	%	Min	%	Min	%	Min	%
3rd	no	2314	(96.0)	2261	(93.8)	2330	(96.7)	6905	(95.5)
	yes	96	(4.0)	149	(6.2)	80	(3.3)	325	(4.5)
6th	no	2257	(93.7)	2244	(93.1)	2306	(95.7)	6807	(94.1)
	yes	153	(6.3)	166	(6.9)	104	(4.3)	423	(5.9)
9th	no	2268	(93.7)	2216	(91.9)	2277	(94.5)	6761	(93.8)
	yes	142	(5.9)	194	(8.1)	133	(5.5)	469	(6.2)
12th	no	2281	(94.6)	2218	(90.5)	2278	(94.5)	6739	(93.5)
	yes	129	(5.4)	230	9.5	132	(5.5)	491	(6.5)
Total	no	9120	(94.6)	8.939	92.4	9191	(95.3)	27,212	(94.1)
	yes	520	(5.4)	739	7.6	449	(4.7)	1708	(5.9)

**Table 2 insects-14-00661-t002:** Glmer model (family = binomial) with the mating frequency as response variable. The response variable was analyzed as event occurrence according to a binomial distribution (0, 1): The applied doses (0, 60, 80 Gy) are compared with the dose 0 (the reference level); the experimental trends (hours) of the experiment are treated as a factor (four levels with the first hour as reference level). The effect of the days is also considered in the analysis (four levels with the third day as reference level). In the case of interaction, the reference level was day 3rd × dose 0 Gy. Significant *p*-values are highlighted in bold.

Fixed Effects:	Estimate	±SE	z Value	*p*-Value
(Intercept)	−2.8855	0.1990	−14.502	<2 × 10^−16^
Dose 60 Gy	0.4658	0.2583	1.803	0.0714
Dose 80 Gy	−0.2703	0.2704	−1.000	0.3174
6th day	0.4994	0.1336	3.739	**0.0002**
9th day	0.4187	0.1354	3.093	**0.0019**
12th day	0.3144	0.1380	2.279	**0.0227**
hour2	−0.3635	0.1148	−3.166	**0.0015**
hour3	−0.6167	0.1181	−5.220	**1.8 × 10^−7^**
hour4	−0.8324	0.1212	−6.871	**6.4 × 10^−12^**
Dose 60 Gy: 6th day	−0.3810	0.1777	−2.144	**0.0320**
Dose 80 Gy: 6th day	−0.2217	0.2019	−1.098	0.2722
Dose 60 Gy: 9th day	0.1282	0.1766	−0.726	0.4681
Dose 80 Gy: 9th day	0.1226	0.1980	0.619	0.5359
Dose 60 Gy: 12th day	0.1682	0.1765	0.953	0.3403
Dose 80 Gy: 12th day	0.2190	0.1999	1.095	0.2734

**Table 3 insects-14-00661-t003:** Glmer model (family = negative binomial) with mating duration in minutes as response variable. The applied doses (0, 60, 80 Gy) are compared with the dose 0 (the reference level); the effect of the days (four levels) is also considered in the analysis (third day as reference level). Significant *p*-values are highlighted in bold.

Fixed Effects	Estimate	±SE	z Value	*p*-Value
(Intercept)	0.999	0.067	14.807	**<2 × 10^−16^**
Dose 60 Gy	0.114	0.057	1.988	**0.0468**
Dose 80 Gy	0.121	0.065	1.862	0.0626
6th day	0.141	0.074	1.903	0.0570
9th day	0.058	0.073	0.805	0.4210
12th day	0.034	0.071	0.483	0.6291

**Table 4 insects-14-00661-t004:** Glmer model (family = negative binomial) with the n. of events per day as response variable. The applied doses (0, 60, 80 Gy) are compared with the dose 0 (the reference level); the effect of the days (four levels) is also considered in the analysis (third day as reference level). Significant *p*-values are highlighted in bold.

Fixed Effects	Estimate	±SE	z Value	*p*-Value
(Intercept)	1.210	0.169	7.166	**7.72 × 10^−13^**
dose60	0.269	0.204	1.316	0.188298
dose80	−0.300	0.213	−1.409	0.158757
6th day	0.136	0.130	1.046	0.295532
9th day	0.276	0.126	2.198	**0.027981**
12th day	0.405	0.122	3.315	**0.000916**

**Table 5 insects-14-00661-t005:** Glmer model (family = negative binomial) with the time elapsed in minutes before the first mating as response variable. The applied doses (0, 60, 80 Gy) are compared with the dose 0 (the reference level); the effect of the day (four levels) is also considered in the analysis. Significant *p*-values are highlighted in bold.

Fixed Effects	Estimate	±SE	z Value	*p*-Value
(Intercept)	3.517	0.389	9.038	<2 × 10^−16^
Dose 60 Gy	−0.966	0.535	−1.806	0.0709
Dose 80 Gy	0.903	0.548	1.646	0.0997
6th day	0.121	0.460	0.262	0.7930
9th day	−0.556	0.476	−1.167	0.2431
12th day	−0.881	0.458	−1.923	0.0545
Dose 60 Gy: 6th day	0.836	0.660	1.266	0.2055
Dose 80 Gy: 6th day	−0.554	0.668	−0.829	0.4072
Dose 60 Gy: 9th day	1.290	0.658	1.961	**0.0499**
Dose 80 Gy: 9th day	−0.330	0.667	−0.495	0.6207
Dose 60 Gy: 12th day	1.323	0.622	2.126	**0.0335**
Dose 80 Gy: 12th day	−0.698	0.655	−1.067	0.2862

**Table 6 insects-14-00661-t006:** Number of minutes spent in mating throughout the entire experiment, with the three possible alternatives: no mating, mating with the untreated male (fertile) and mated with the irradiated male. In the control, both males are untreated but are still marked to distinguish their behaviors, similar to the experimental set with irradiated males. Total means are calculated only on the experiments at 60 and 80 Gy.

Minutes Elapsed				
Treatment	Dose (Gy)			Total Mean
	0	60	80	
No mating	902.3	888.2	873.6	877.9
“fertile”	28.5	24.5	29.8	27.2
“irradiated”	29.2	47.3	56.6	51.9
Total	960.0	960.0	960.0	960.0

**Table 7 insects-14-00661-t007:** Glmer model (family = negative binomial): response variable was analyzed as number of minutes spent in copulation per day: The applied treatment is nested into the applied doses (0, 60, 80 Gy) and the model estimates are referred to the zero hypothesis of no difference between irradiated and fertile. The effect of the days (four levels) is considered in the analysis (third day as reference level). The males at the dose zero were used as a further control. Significant *p*-values are highlighted in bold.

Fixed Effects	Estimate	±SE	z Value	*p*-Value
(Intercept)	1.2842	0.2032	6.319	**2.64 × 10^−10^**
6th day	0.7007	0.1956	3.583	**0.00034**
9th day	0.5744	0.1966	2.922	**0.00348**
12th day	0.0329	0.1993	0.165	0.86885
treatm: dose 0 Gy	0.1895	0.2706	0.700	0.48380
treatm: dose 60 Gy	0.7599	0.2697	2.818	**0.00484**
treatm: dose 80 Gy	0.8177	0.2678	3.054	**0.00226**

**Table 8 insects-14-00661-t008:** Means and standard errors of the time spent in copulation in function of the applied dose of irradiation and the day and hour from the beginning of the experiment. In the case of the hours, the mating frequency is summed across the four days of the experiment. The male with the subscript “2” is the irradiated male, except in the case of the dose zero where both the males are untreated.

Dose	Days	Male_(1)_		Male_(2)_		Hours	Male_(1)_		Male_(2)_	
		(Mins)	±SE	(Mins)	±SE		(Mins)	±SE	(Mins)	±SE
0	3	6.60	2.47	2.90	1.13	1	7.80	2.14	10.90	2.05
	6	10.60	2.87	13.60	3.56	2	9.40	2.95	7.50	1.81
	9	8.30	2.09	6.60	1.92	3	5.50	1.55	6.30	1.37
	12	3.00	1.27	6.10	2.36	4	5.80	2.02	4.50	1.20
	mean	7.13	1.17	7.30	1.32	mean	7.13	1.10	7.30	0.87
60	3	2.80	1.19	10.80	3.82	1	9.60	3.63	16.30	4.58
	6	7.80	3.06	14.00	3.43	2	5.90	1.87	13.70	2.50
	9	8.80	1.90	10.20	2.76	3	5.30	1.22	8.50	1.93
	12	5.10	1.52	12.30	2.41	4	3.70	1.11	8.80	2.27
	mean	6.13	1.05	11.83	1.53	mean	6.13	1.11	11.83	1.54
80	3	7.20	3.78	11.90	2.98	1	11.40	2.85	23.80	5.68
	6	8.10	2.01	18.10	5.26	2	5.40	2.24	12.30	2.83
	9	10.60	2.15	15.20	2.81	3	8.70	2.63	11.20	2.79
	12	3.90	2.47	11.40	2.90	4	4.30	1.30	9.30	1.90
	mean	7.45	1.35	14.15	1.80	mean	7.45	1.21	14.15	1.95
Total means *	3	5.53	1.55	11.35	2.36	1	9.60	1.66	20.05	3.65
	6	8.83	1.52	16.05	3.09	2	6.90	1.38	13.00	1.84
	9	9.23	1.16	12.70	2.00	3	6.50	1.10	9.85	1.68
	12	4.00	1.03	11.85	1.84	4	4.60	0.87	9.05	1.44
	mean	6.90	0.69	12.99	1.18	Total	6.90	0.66	12.99	1.18

* In the case of male 2, the total means are computed only for the mating with irradiated individuals.

**Table 9 insects-14-00661-t009:** Glmer model (family = negative binomial): the response variable was analyzed as number of minutes of mating per hour: The applied treatment is nested into the experimental doses (0, 60 and 80 Gy) and the model estimates are referred to the zero hypothesis of no difference in mating frequencies between irradiated and fertile; the males at the dose zero were both not irradiated and were used as a further control. The experimental trend (hours) of the experiment is treated as a factor (four levels, first hour as reference level). Significant *p*-values are highlighted in bold.

Fixed Effects:	Estimate	±SE	z Value	*p*-Value
(Intercept)	2.003	0.166	12.075	**<2 × 10^−16^**
h2	−0.386	0.126	−3.104	**0.001910**
h3	−0.487	0.127	−3.834	**0.000126**
h4	−0.756	0.129	−5.846	**5.05 × 10^−9^**
treatm: dose 0 Gy	0.202	0.268	0.752	0.451903
treatm: dose 60 Gy	0.712	0.267	2.667	**0.007645**
treatm: dose 80 Gy	0.753	0.266	2.832	**0.004619**

**Table 10 insects-14-00661-t010:** Glmer model (family = negative binomial) with the duration of mating events as outcome variable. The response variable was analyzed as duration of event in minutes: The treatment is nested into the applied doses (0, 60 and 80 Gy) and the model estimates are referred to the zero hypothesis of no difference in mating duration between irradiated and fertile. The males at the dose zero were both not irradiated and were used as a further control. Significant *p*-values are highlighted in bold.

Fixed Effects	Estimate	SE	z Value	*p*-Value
(Intercept)	1.1123	0.0354	31.465	**<2 × 10^−16^**
treatm: dose 0 Gy	0.0130	0.0676	0.193	0.8472
treatm: dose 60 Gy	0.1115	0.0581	1.919	0.0549
treatm: dose 80 Gy	0.0291	0.0559	0.521	0.6025

**Table 11 insects-14-00661-t011:** Means and standard errors of the response variables duration of mating in minutes, mating events per day and minutes elapsed before the first mating, in function of the applied dose of irradiation and the day from the beginning of the experiment. The male with the subscript “2” is the irradiated male, except in the case of the dose zero where both the males are untreated. The dash indicates no data available.

		Duration of Mating (Min Day^−1^)				Mating Events (n. Day^−1^)				Time Elapsed before the 1st Mating (Min)			
Dose	Days	Male_(1)_	±SE	Male_(2)_	±SE	Male_(1)_	±SE	Male_(2)_	±SE	Male_(1)_	±SE	Male_(2)_	±SE
0	3	3.47	0.39	2.90	0.38	1.90	0.71	1.00	0.39	108.0	-	1.50	0.50
	6	3.03	0.18	3.02	0.22	3.50	0.82	4.50	1.19	35.00	10.51	28.50	16.6
	9	2.96	0.22	3.24	0.42	2.80	0.63	2.10	0.59	19.75	7.90	16.00	6.52
	12	3.00	0.30	3.15	0.28	1.00	0.42	2.00	0.63	18.00	6.00	13.00	4.36
	mean	3.10	0.13	3.08	0.15	2.30	0.35	2.40	0.42	33.17	8.64	16.86	5.42
60	3	2.55	0.31	3.45	0.35	1.10	0.43	3.10	1.07	-	-	16.50	2.50
	6	3.39	0.45	3.50	0.27	2.30	0.80	4.00	0.98	18.50	17.50	3.00	2.00
	9	3.14	0.30	3.81	0.39	2.80	0.51	2.70	0.56	32.00	16.59	3.50	0.50
	12	3.00	0.19	3.00	0.19	1.70	0.50	4.10	0.74	-	-	21.29	10.8
	mean	3.10	0.18	3.40	0.15	1.97	0.30	3.48	0.42	28.63	12.78	15.00	6.09
80	3	3.00	0.17	3.16	0.32	2.40	1.29	3.70	0.94	58.67	56.67	-	-
	6	3.52	0.42	3.55	0.18	2.30	0.58	5.10	1.35	47.80	37.47	3.00	0.58
	9	2.79	0.17	2.92	0.15	3.80	0.80	5.20	0.87	10.00	3.00	7.80	2.65
	12	2.44	0.13	2.82	0.16	1.60	0.98	4.00	0.97	4.00	-	3.33	0.33
	mean	2.95	0.13	3.13	0.10	2.53	0.47	4.50	0.52	37.42	19.98	5.27	1.36
Total mean *	3	3.07	0.17	3.35	0.50	1.80	0.50	3.40	0.70	71.00	41.93	16.50	2.50
	6	3.27	0.19	3.69	0.37	2.70	0.43	4.55	0.82	37.58	15.62	3.00	0.71
	9	2.95	0.13	3.14	0.19	3.13	0.38	3.95	0.58	23.15	8.04	6.57	2.00
	12	2.79	0.12	3.17	0.24	1.43	0.38	4.05	0.59	13.33	5.81	15.90	7.90
	mean	3.04	0.08	3.33	0.16	2.27	0.22	3.46	0.34	33.62	8.52	10.54	3.45

* In the case of male 2, the total means of doses are computed only for the mating with irradiated individuals.

**Table 12 insects-14-00661-t012:** Glmer model (family = negative binomial) with the number of mating events per day as outcome variable. The applied treatment is nested into the applied doses (0, 60 and 80 Gy) and the model estimates are referred to the zero hypothesis of no difference between irradiated and fertile. The males at the dose zero were both not irradiated and were used as a further control. The effect of the days is included in the model (third day as reference level). Significant *p*-values are highlighted in bold.

Fixed Effects:	Estimate	SE	z Value	*p*-Value
(Intercept)	−0.0366	0.2291	−0.160	0.87303
6th day	0.5306	0.1585	3.348	**0.00081**
9th day	0.4429	0.1598	2.772	**0.00557**
12th day	0.0566	0.1654	0.342	0.73210
treatm: dose 0 Gy	0.2704	0.1420	1.905	0.05683
treatm: dose 60 Gy	0.4835	0.1382	3.497	**0.00047**
treatm: dose 80 Gy	0.5722	0.1346	4.252	**2.11 × 10^−5^**

**Table 13 insects-14-00661-t013:** Glmer model (family = binomial) with the first mating occurrence as outcome variable. The response variable was analyzed event occurrence (0, 1): The applied treatment is nested into the applied doses (0, 60, 80 Gy) and the model estimates are referred to the zero hypothesis of no difference in mating occurrence between irradiated and fertile males. The males at the dose zero were both not irradiated and were used as a further control.

Fixed Effects	Estimate	SE	z Value	*p*-Value
Dose 0 Gy	0.2089	0.5431	0.385	0.7005
Dose 60 Gy	0.9740	0.5598	1.740	0.0819
Dose 80 Gy	0.5459	0.5339	1.022	0.3066

**Table 14 insects-14-00661-t014:** Glmer model (family = negative binomial) with the time elapsed before the first mating as outcome variable. Data were selected when both males mated once at least. The applied treatment is nested into the applied doses (0, 60 and 80 Gy) and the model estimates are referred to the zero hypothesis of no difference between irradiated and fertile. The males at the dose zero were both not irradiated and were used as a further control. Significant *p*-values are highlighted in bold.

Fixed Effects	Estimate	SE	Z Value	*p*-Value
(Intercept)	3.2517	0.2305	14.110	**<2 × 10^−16^**
treatm:dose 0 Gy	−0.5975	0.3738	−1.599	0.109918
treatm:dose 60 Gy	−0.6204	0.3965	−1.564	0.117722
treatm:dose 80 Gy	−1.5798	0.4351	−3.631	**0.000282**

## Data Availability

The data presented in this study are available on request from the corresponding author.

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
