# Peer review of "Effects of γ-Irradiation on Mating Behavior of Red Palm Weevil, *Rhynchophorus ferrugineus* (Olivier, 1790) (Coleoptera: Dryophthoridae)"

_insects, 2023, doi:10.3390/insects14070661_

Round 1
Reviewer 1 Report
This study evaluates the sexual competitiveness of irradiated R. ferrugineus males in matings competing against fertile males, both under non-choice and choice conditions (under a promiscuous context) over 12 days. The authors profusely present their results with adequate statistical analyses, showing that the irradiated doses (60 and 80 Gy) did not affect the sexual competitiveness of the males and even the males irradiated at 80 gy performed better with respect to the number of mating achieved throughout the experimental time. This undoubtedly represents a step forward in the long road of evaluating the feasibility of using the SIT against this pest.
I believe that the objectives were adequately achieved, and that this MS has enough merit to be published in Insects. Here are some comments that authors need to respond/address before being accepted for publication.
Lines 40-41, these lines are out of context, please delete it.
Lines 146-155, collected field individuals were measured in some way (ie, size in mm, weight in mg) to determine the homogeneity/diversity of the field population and whether any phenotype (eg, size) was more successful than others. In some species larger males are more competitive.
Lines 497-500, I agree with what the authors state in these lines, but for the success of the SIT the refractory period of the females after the first mating is also a crucial issue, and in R. ferrugineus females this characteristic is completely absent. Although the authors present valid arguments in lines 113-119 of the Introduction, I consider this topic deserves a deep discussion.
References
There are several typos along the cited references, some of them are incomplete.
Author Response
Authors: We are very grateful for the work of the anonymous reviewer for his/her detailed and valuable comments that really helped to improve the quality our manuscript.
We accepted all the suggestions from the reviewer, providing our point of view in the file.
Reviewer 1
This study evaluates the sexual competitiveness of irradiated R. ferrugineus males in matings competing against fertile males, both under non-choice and choice conditions (under a promiscuous context) over 12 days. The authors profusely present their results with adequate statistical analyses, showing that the irradiated doses (60 and 80 Gy) did not affect the sexual competitiveness of the males and even the males irradiated at 80 gy performed better with respect to the number of mating achieved throughout the experimental time. This undoubtedly represents a step forward in the long road of evaluating the feasibility of using the SIT against this pest.
I believe that the objectives were adequately achieved, and that this MS has enough merit to be published in Insects. Here are some comments that authors need to respond/address before being accepted for publication.
Lines 40-41, these lines are out of context, please delete it.
Authors: Ok, we deleted the lines as suggested by the reviewer.
Lines 146-155, collected field individuals were measured in some way (ie, size in mm, weight in mg) to determine the homogeneity/diversity of the field population and whether any phenotype (eg, size) was more successful than others. In some species larger males are more competitive.
Authors: as shown in the work of Inghilesi et al. (2014) and Mazza et al. (2015), differences in sexual performance among males were not related to male size and irradiation status. We have clarified this better at lines 203-207.
Lines 497-500, I agree with what the authors state in these lines, but for the success of the SIT the refractory period of the females after the first mating is also a crucial issue, and in R. ferrugineus females this characteristic is completely absent. Although the authors present valid arguments in lines 113-119 of the Introduction, I consider this topic deserves a deep discussion.
Authors: We agree that this topic is important, so we have added some lines about this relevant aspect in the Discussion section, at the lines 659-664 of the new version of the manuscript.
References: There are several typos along the cited references, some of them are incomplete.
Authors: We corrected all the reference section.
Reviewer 2 Report
Summary:
In this article, the authors test the mating behavior of irradiated male red palm weevil (at two different sterilization doses) in caged studies either alone with a female, or when competing with a wild male. This work addresses several important aspects of mating behavior, and the majority of findings show that red palm weevil males were not significantly negatively impact by radiation. In fact, it appears that there may be some hermetic effects induced, seeing irradiated males spending more time mating and mating more times during assays. Overall, this is an interesting and unique look at a critical factor when establishing a sterile insect technique program for a pest species.
General Comments:
While this work is appropriately researched and the methods are sound overall, there are several key aspects not mentioned that are integral to sterile insect technique that need to be addressed by the authors. Primarily, there is no information on the age of the insects used in this study. Insect age is known to have a large impact on response to radiation dose as well as mating behavior. Both males and females need to be age-controlled in these experiments, as age differences can greatly impact observations during mating assays. Many other areas where more information is necessary can be found in the line-by-line comments.
Additionally, the tables and figures are generally good, but many figures would be greatly improved by including the letter groupings from Tukey’s HSD tests to show significant differences among treatments. This would greatly increase the reader’s understanding of the findings.
Comments by line
59-64: This sentence is disorganized and difficult to follow. I would suggest that it be rewritten as two sentences, first describing the economic, environmental, and human harms. Then follow up with the landscape and biodiversity harms, including the palm species at risk.
93-94: I would suggest qualifying this statement that “whose ability to mate is minimally compromised” because although the goal is ideally not to compromise, in practice there are usually some negative impacts.
110-112: This sentence should be the end of the previous paragraph and not stand alone.
120-123 and 128-132: Both paragraphs are essentially the same statement. The latter works better, I would eliminate the first.
151-155: More information is needed. Were the field-collected individuals adults or larvae? How many weevils were reared in each cage? Were sexes kept separate during rearing?
156-159: What was the age of the adult males when they were irradiated? It is well documented that insect age can have a large impact on both dose required to achieve sterilization as well as the occurrence of negative physiological effects. Additionally, were any of the weevils in this study assessed for sterility?
161-162: How were females ensured to have not mated before any experiments? Also, what was the size of the cages used in mating assays? Were all insects virgin before mating assays?
165: In choice assays, how did you distinguish the irradiated males from the wild males?
171-174: How were adults kept (cage type? Alone or grouped?) prior to mating and were males and females added to mating cages at the same time? Also, is there biological significance to mating 3, 6, 9, and 12 days after irradiation? Again, insect age is critically important to mention. If males were not the same age across groups, that makes results unreliable.
177: Specify “visual observations”, or were the weevils recorded with a camera?
187: Size of the jars females were isolated in? Also, why were females isolated after mating? There does not appear to be any further data collection after mating.
Data analysis: Somewhere in here, mention that you also looked at interactions across fixed variables. I see that interactions are reported in some of the results, but not all. If they were all tested for interactions, but are only reported if interactions were significant, this information belongs in the analysis section.
234: In this entire section I believe you are looking at total time spent mating, which is not frequency. Frequency would be the overall number of mating events in the span of 4 hours. This appears to be the number of minutes weevils were observed as mating or not.
Table 1: For the total of 60 Gy at 0 mating, you have a decimal point that does not belong (8.939).
249: You looked at an interactions between dose*day and dose*hour, but not hour*day or dose*hour*day? Was the reference for the GLmer in table 2 0 Gy/Hour 1/day 3?
269: Here you say, “the first and second day of the experiment”, do you mean day 3 and day 6 after irradiation? This is confusing.
270: Does this refer to figure 2-B?
302-311: Did you check for interactions in this analysis?
343-347: This is confusing. Did you include males that never mated in the analysis? If they did not mate, you could not record the elapsed time before the firs mating. It looks like this statement is saying that if males dosed to 80 Gy did mate, they mated earlier than 0 or 60 Gy, but that the ones that did not mate were affecting the average. If non-mating males were included in this analysis, it needs to be redone without the males that did not mate.
353-356: Again, there is confusion when you say frequencies. The frequency of mating would be the total number of mating events, recorded as an integer. This looks like total time mating across 4 hours.
381: This should say “irradiated males mated for longer durations than untreated ones” not “more frequently”.
384: Again, you mention “day 2”, is this really day 6 after irradiation? Base all time points as days after irradiation to prevent confusion.
442-448: I think this is possibly the most important finding in this study, and it deserves more attention. What you are saying is that irradiated males mate more times than wild ones (granted, in a cage study). If this trend holds up in the field, sterilized males could inseminate multiple females and out compete wild males. You mention in the introduction that this weevil has last male precedence, so it would be interesting if you had data on the last male observed mating in choice assays to potentially strengthen this argument.
453: Fig 7-B tells the story! But add (Gy) to the x axis label.
477: Fig 8-A is missing standard error bars.
524-526: This is hard to claim in a lab study where males and females are caged in close quarters. I think you are overstating by saying your findings “exclude the possibility of that radiation significantly interferes with male insect’s physiology”.
547-550: While I agree that number of matings is an indication of increased competitiveness, what is important about first mating in a species that has last male precedence?
577-580: I would rephrase this to say “this work has focused on using sterilizing radiation doses while maintaining the mating competitiveness…” as this work did not focus on the selection of doses or looking at the hatching of eggs from mated females in this study.
649: I think this table is needed in the main manuscript, to show data on number of mating events.
The majority of language issues were minor, with the exception of the use of the term “frequencies”. I mention this in the review comments, but it is very important to clarify that frequency is defined as the number of events in a unit of time.
23: no need for “it” in “30 years and it is responsible for…”
38: change “can make” to “provide”
54: change to- “in areas of recent colonization populations of RPW have stabilized”
74: Remove “the” immediately before “RPW”
76: Say “Once the palm shows the first symptoms”, not “when”.
163: “(60 or 80 Gy)” not “and”
329: “no differences were found” not “not”
396: “Regarding” not “as regards”
523: “clearly confirmed that neither of the two” not “none of the two”
531: “was not significantly different” instead of “was significantly not different”
578: “radiation dose” instead of “irradiation dose”
Author Response
Authors: We are very grateful for the work of the anonymous reviewer for his/her detailed and valuable comments that really helped to improve the quality our manuscript.
We accepted all the suggestions from the reviewer, providing our point of view in the file.
Reviewer #2
Summary:
In this article, the authors test the mating behavior of irradiated male red palm weevil (at two different sterilization doses) in caged studies either alone with a female, or when competing with a wild male. This work addresses several important aspects of mating behavior, and the majority of findings show that red palm weevil males were not significantly negatively impact by radiation. In fact, it appears that there may be some hermetic effects induced, seeing irradiated males spending more time mating and mating more times during assays. Overall, this is an interesting and unique look at a critical factor when establishing a sterile insect technique program for a pest species.
General Comments:
While this work is appropriately researched and the methods are sound overall, there are several key aspects not mentioned that are integral to sterile insect technique that need to be addressed by the authors. Primarily, there is no information on the age of the insects used in this study. Insect age is known to have a large impact on response to radiation dose as well as mating behavior. Both males and females need to be age-controlled in these experiments, as age differences can greatly impact observations during mating assays.
Authors: We really appreciated this comment. We fully agree with the reviewer about this point: several studies demonstrated that the age of medfly males is extremely important for a correct application of SIT on that pest species. However, red palm weevil is very complicated to rear in captivity (long life cycle, presence of cannibalism at the larval stages, needs of correct material and climatic conditions to spin a cocoon around the prepupa stage suitable to complete the metamorphosis) and has a different biology. Moreover, in previous studies (Inghilesi et al, 2014; Musmeci et al., 2018), behavioral studies and last mating male sperm precedence observations have been carried out on adults (males and females) collected in the field (mainly on rhyncho-traps baited with the aggregation pheromone). The results of the work done by Inghilesi et al. (2014) were confirmed one year later by Mazza et al. (2015), with newly emerged adults (fertile and irradiated), showing that, in gregarious conditions, males -independently if they were irradiated or fertile, newly emerged or old) were behaving in the same way: females, regardless to the physiological status (virgin or already mated, newly emerged of collected in the field) were always showing a passive pattern; vice versa, (most of the) males were extremely sexually actives, showing even a very promiscuous mating behavior. In Inghilesi et al. (2014) and Mazza et al. (2015) papers, the differences in sexual performance among the males were not related with the size of the males and to the irradiation status.
The experimental design of the paper Musmeci et al., 2018 was built to demonstrate the presence of the last mating male sperm precedence mechanism: at the beginning (first two experiments) using newly emerged virgin adults and later (to simulate what would be the “real situation in the field”) a third experiment was set up using already mated adults (both females and males): in these third experiment a field collected female was left several days alone in a cage, recording oviposition and larval development. After 9 days an irradiated (field collected) male was introduced in the cage: the fertile female continued to lay eggs, but all of them were sterile.
We do agree with the reviewer that -in the first version of the manuscript- we did not spend too much effort explaining the reasons why we used field collected adults. Even if the work done was addressed mainly to evaluate the sexual performance of sterile males (in no-choice and choice conditions), we agree that it is important to provide additional background data on how this work can provide additional inputs on the possibility to include SIT in an Area-Wide multi-disciplinary management strategy to suppress this important pest species. For this reason, we enhance the manuscript with additional information (explained more in details in the line-by-line comments).
Many other areas where more information is necessary can be found in the line-by-line comments.
Additionally, the tables and figures are generally good, but many figures would be greatly improved by including the letter groupings from Tukey’s HSD tests to show significant differences among treatments. This would greatly increase the reader’s understanding of the findings.
Authors: We agree with the reviewer about Tukey’s test and letters. However, when there are interactions, the post-hoc estimates are problematic especially for non-normal data and when sample size is not very large. Moreover, the comparisons between 2 means, when possible and useful to the description of results, are reported in the text. In any case, the s.e. is always reported in the figures and should help to estimate the magnitude of statistical uncertainty (a double s.e. is quite close to 95% CI).
Comments by line
59-64) This sentence is disorganized and difficult to follow. I would suggest that it be rewritten as two sentences, first describing the economic, environmental, and human harms. Then follow up with the landscape and biodiversity harms, including the palm species at risk.
Authors: We changed this sentence as suggested.
93-94) I would suggest qualifying this statement that “whose ability to mate is minimally compromised” because although the goal is ideally not to compromise, in practice there are usually some negative impacts.
Authors: We have made the suggested change.
110-112) This sentence should be the end of the previous paragraph and not stand alone.
Authors: We have made the suggested change.
120-123 and 128-132: Both paragraphs are essentially the same statement. The latter works better, I would eliminate the first.
Authors: We have made the suggested change.
151-155: More information is needed. Were the field-collected individuals adults or larvae? How many weevils were reared in each cage? Were sexes kept separate during rearing?
Authors: As the reviewer suggested we have expanded this paragraph and made it clearer
156-159: What was the age of the adult males when they were irradiated? It is well documented that insect age can have a large impact on both doses required to achieve sterilization as well as the occurrence of negative physiological effects. Additionally, were any of the weevils in this study assessed for sterility?
Authors: We replied to this question providing some changes in the text: in the introduction, we added a part (on the new version of the manuscript, see lines 112-119); and we also did some changes in the chapter Materials and Methods (on the new version of the manuscript, lines 146-150).
161-162: How were females ensured to have not mated before any experiments? Also, what was the size of the cages used in mating assays? Were all insects virgin before mating assays?
Authors: As we mentioned before, all the individuals (males and females) were not virgin and already mated (they were collected in field traps). Since previous studies on red palm weevil (Musmeci et al. 2012; Inghilesi et al. 2014; Musmeci et al. 2018) already demonstrated that -for this polyandric species- (differently from Mediterranean fruit fly) the female does not change the sexual behavior when she has the spermathecae full of previous male(s) sperms, and males are always showing a clear active role in the mating (independently if they already mated), we decided to perform the experiments using field collected adults: from our point of view, these conditions are more similar to what would be the real situation in the field, when an irradiated adult male will find an already mated female. Also, in Musmeci et al. 2012, most of the females found in the traps (baited with the aggregation pheromone, emitted by the males) or in the dead palms, were already mated. We agree with the reviewer that additional information was needed, and we improved the text in the Introduction and in Material & Methods sections (see our previous reply referred to the lines 112-119 and 146-150).
165: In choice assays, how did you distinguish the irradiated males from the wild males?
Authors: we modified this point in the manuscript to make it clearer.
171-174: How were adults kept (cage type? Alone or grouped?) prior to mating and were males and females added to mating cages at the same time? Also, is there biological significance to mating 3, 6, 9, and 12 days after irradiation? Again, insect age is critically important to mention. If males were not the same age across groups, that makes results unreliable.
Authors: We agree with the questions, and we provided the changes in the text (see lines 146-150; 151-157; 175-182)
177: Specify “visual observations”, or were the weevils recorded with a camera?
Authors: The observations were visual. We added this detail as suggested by the reviewer.
187: Size of the jars females were isolated in? Also, why were females isolated after mating? There does not appear to be any further data collection after mating.
Authors: females were isolated from males to prevent mating from occurring in the time between observations. We have added details about this part in the text to make it clearer.
Data analysis: Somewhere in here, mention that you also looked at interactions across fixed variables. I see that interactions are reported in some of the results, but not all. If they were all tested for interactions, but are only reported if interactions were significant, this information belongs in the analysis section.
Authors: we reported this in the paragraph on data analysis: “The model design was chosen based on the optimal parsimony principle (AIC and BIC estimators) and on significance of the overall effects.”
234: In this entire section I believe you are looking at total time spent mating, which is not frequency. Frequency would be the overall number of mating events in the span of 4 hours. This appears to be the number of minutes weevils were observed as mating or not.
Authors: we agree with you, frequency could be unclear: we changed it now to: minutes of mating.
Table 1: For the total of 60 Gy at 0 mating, you have a decimal point that does not belong (8.939).
Authors: done.
249: You looked at an interactions between dose*day and dose*hour, but not hour*day or dose*hour*day? Was the reference for the GLmer in table 2 0 Gy/Hour 1/day 3?
Authors: we did not add these interactions in model design on the basis of AIC and BIC criterion and significance (see data analysis section)
Was the reference for the Glmer in table 2 0 Gy/Hour 1/day 3?
Authors: Yes, correct: we added this information in the caption of Table2.
269: Here you say, “the first and second day of the experiment”, do you mean day 3 and day 6 after irradiation? This is confusing.
Authors: We agree with the reviewer, for homogeneity and consistency what we had previously called observation 1, 2, 3 have been changed to 3rd, 6th, 9th day of the experiment throughout the text. The error pointed out by the reviewer was an oversight left in the text.
270: Does this refer to figure 2-B?
Authors: Yes, we added it in the text
302-311: Did you check for interactions in this analysis?
Authors: Yes, we did
343-347: This is confusing. Did you include males that never mated in the analysis? If they did not mate, you could not record the elapsed time before the firs mating. It looks like this statement is saying that if males dosed to 80 Gy did mate, they mated earlier than 0 or 60 Gy, but that the ones that did not mate were affecting the average. If non-mating males were included in this analysis, it needs to be redone without the males that did not mate.
Authors: Yes, we agree, this part can create confusion. We deleted this part now, also according to the suggestions of the reviewer 2 (not so relevant information due to borderline significance). (In the new version, lines 360-363).
353-356: Again, there is confusion when you say frequencies. The frequency of mating would be the total number of mating events, recorded as an integer. This looks like total time mating across 4 hours.
Authors: we agree with you, we changed period in: “The total time mating across the entire duration of the experiment was longer in the irradiated males than the in the fertile ones” (see new lines 369-370).
381: This should say “irradiated males mated for longer durations than untreated ones” not “more frequently”.
Authors: we changed the phrase in: “The irradiated males mated for longer time than untreated ones” (See lines 395-396).
384: Again, you mention “day 2”, is this really day 6 after irradiation? Base all time points as days after irradiation to prevent confusion.
Authors: As answered before, we agree with the reviewer and have corrected these oversights throughout the manuscript.
442-448: I think this is possibly the most important finding in this study, and it deserves more attention. What you are saying is that irradiated males mate more times than wild ones (granted, in a cage study). If this trend holds up in the field, sterilized males could inseminate multiple females and out compete wild males. You mention in the introduction that this weevil has last male precedence, so it would be interesting if you had data on the last male observed mating in choice assays to potentially strengthen this argument.
Authors: We agree with the reviewer that this aspect is very important and interesting, but unfortunately, we did not collect data on fertility, as our study was intended to focus only on behavior. However, many studies confirming the last male sperm precedence have already been done (see Musmeci et al. 2018) and we have specified it more in the text at the lines 112-119; 601-607.
453: Fig 7-B tells the story! But add (Gy) to the x axis label.
Authors: we corrected the figure as suggested by reviewer.
477: Fig 8-A is missing standard error bars.
Authors: There is no standard error bar because the data indicate a number of events (total frequency).
524-526: This is hard to claim in a lab study where males and females are caged in close quarters. I think you are overstating by saying your findings “exclude the possibility of that radiation significantly interferes with male insect’s physiology”.
Authors: We were not correct: we made some changes in the paragraph (see the new lines 546-553).
547-550: While I agree that number of matings is an indication of increased competitiveness, what is important about first mating in a species that has last male precedence?
Authors: The question would be correct, if we were dealing with monogamous males. But in the case of red palm weevil, where (already mated) females are always showing a passive mating behavior and males are actively competing among each other for the mating, the presence of this highly competitive performance of the irradiated males is another good point for their possible success in open field conditions. To avoid confusion, we improved the discussion with more information (see new lines 574-580).
577-580: I would rephrase this to say “this work has focused on using sterilizing radiation doses while maintaining the mating competitiveness…” as this work did not focus on the selection of doses or looking at the hatching of eggs from mated females in this study.
Authors: We agree. We also decided to move the full paragraph in the Conclusions Chapter.
649: I think this table is needed in the main manuscript, to show data on number of mating events.
Authors: Okay, we moved the Table to the main manuscript.
Comments on the Quality of English Language
The majority of language issues were minor, with the exception of the use of the term “frequencies”. I mention this in the review comments, but it is very important to clarify that frequency is defined as the number of events in a unit of time.
Authors: we made the change suggested by the reviewer.
23: no need for “it” in “30 years and it is responsible for…”
Authors: Ok, we made the suggested correction.
38: change “can make” to “provide”
Authors: Ok, we made the suggested correction.
54: change to- “in areas of recent colonization populations of RPW have stabilized”
Authors: Ok, we made the suggested correction.
74: Remove “the” immediately before “RPW”
Authors: Ok, we made the suggested correction
76: Say “Once the palm shows the first symptoms”, not “when”.
Authors: Ok, we made the suggested correction
163: “(60 or 80 Gy)” not “and”
Authors: Ok, we made the suggested correction
329: “no differences were found” not “not”
Authors: Ok, we made the suggested correction
396: “Regarding” not “as regards”
Authors: Ok, we made the suggested correction
523: “clearly confirmed that neither of the two” not “none of the two”
Authors: Ok, we made the suggested correction
531: “was not significantly different” instead of “was significantly not different”
Authors: Ok, we made the suggested correction
578: “radiation dose” instead of “irradiation dose”
Authors: Ok, we made the suggested correction
Reviewer 3 Report
Cristofaro et al. report the results of two laboratory experiments on the mating behavior of red palm weevil (RPW) in the presence of irradiated males. The study is relevant to the use of sterile males for SIT based pest control.
MAIN POINTS:
The manuscript is very confusing. The use of GLM models takes precedence over the biological processes being analyzed.
GLMs are a useful tool, but in the manner in which they are reported in the manuscript, they confuse and obscure the results rather than proving a clear analysis of main effects and interactions.
The Introduction is too long and should focus on the use of SIT against the RPW.
The Summary and Abstract should focus on the results and their relevance to SIT.
The clarity of the Results section requires major improvement.
I have written suggestions and numbered points on a scanned copy of the manuscript.
NUMBERED POINTS (see scanned manuscript)
1. 60% of the simple summary is preamble. Please focus on the results and their meaning.
2. Ditto for the Abstract.
3. Two long paragraphs on the importance of RPW as a pest. Please reduce it to a few lines, we understand that it's an important pest.
4. Please focus on why it is difficult to control by conventional means.
5. The key to the SIT is the release of MASSIVE numbers of sterile males, so that the majority of matings of wild females occur with sterile males. This is even more important in females remate.
6. I disagree. If females have "an equal chance" of receiving sterile and fertile sperm the SIT technique will be extremely inefficient! If females have a 10:1 chance or more of receiving sterile sperm, the technique will probably be more effective.
7. Should be "Cobalt-60 rays"
8. Age of males when irradiated?, age when tested? It was unclear to me whether the insects were reared through several generations in the laboratory or not?
9a. Glazes? Do you mean paint? What type of paint?
9b. What time of day were observations performed? It seems from the results that mating declined over time, so this is important to clarify.
10. fecundation? You mean insemination?
11. Table 1 indicates the number of mating events, correct? Please state this explicitly. No need to use codes (0,1) that you employed in your statistical analysis.
12. Percentage values can be given in parentheses beside the number of matings.
13. Why do you use asterisks in the text? Delete them.
14. Please indicate degrees of freedom for Chi2 values throughout the manuscript.
15. Why give asterisks and points for P-values? If you wish to highlight significant terms, present the p-values in bold (normal text for non-significant values).
16. Tendency is not the correct term. You mean the effect was borderline significant (i.e. P = 0.1-0.05).
17. What point are you making here? This text seems vague. Is this an issue that should be discussed in the Discussion?
18. Table 6 title is difficult to understand. Title says n of observations per minute, but Table says count in minutes (as if it were a duration). Not a count I think.
19. Table 8. Frequency of matings in minutes??? I find it hard to believe that insects were mating 2 to 10 times per minute! Table text says minutes/day. Sounds more like a duration of mating to me, i.e. "time spent in copulation". Again, this needs to be made clear.
20. Means of doses?? This sounds incorrect.
21.a. What is the correct y-axis label for Fig 5?
21b. Table 9. Mating frequency in number of minutes per hour? This makes no sense.
22. Figure 6. Has the same y-axis issue as Fig 5.
23. Higher than what?
24. This paragraph is confusing.
25. No significant differences in what exactly?
26. I did not understand the importance of this text. Is this something that needs to be moved to the Discussion?
27. Surely a simple contingency table or Fisher's exact test would provide a clear analysis of this frequency data, rather than analyzing proportions (using the GLM model).
28. Please clarify the y-axis labels on Fig 8a and 8b.
29. Lower than what?
30. See my comment 5; SIT is based on the propensity of fertile females to mate with sterile males which greatly outnumber fertile males.
31. What bottleneck?
32. Drastic shifts in what?
33. Unclear whether laboratory rearing was performed over several generations or not, although this text indicates insects were wild-caught (and used immediately?).
34. If RPW males were reared over several generations in the laboratory they will have been selected to mate rapidly with females, as happens in mass-reared tephritid flies.
35. You do not seem to consider the insemination capacity of irradiated males. i.e. attempted matings vs. successful matings.
36. This is a long text that should focus tightly on the results of the study. Please rewrite and reduce to two or three sentences, as your findings are simple and do not require a complex text.
37. The titles of the tables in the annex can be improved for clarity. I was confused by Table A4.
38. Also confused by Table A6.
39. Information is missing in the references (I only checked the first page).

Needs editing.
Author Response
Authors: We are very grateful for the work of the anonymous reviewer for his/her detailed and valuable comments that really helped to improve the quality our manuscript.
We accepted all the suggestions from the reviewer, providing our point of view in the file.
Reviewer #3
Comments and Suggestions for Authors
Cristofaro et al. report the results of two laboratory experiments on the mating behavior of red palm weevil (RPW) in the presence of irradiated males. The study is relevant to the use of sterile males for SIT based pest control.
MAIN POINTS:
The manuscript is very confusing. The use of GLM models takes precedence over the biological processes being analyzed.
GLMs are a useful tool, but in the manner in which they are reported in the manuscript, they confuse and obscure the results rather than proving a clear analysis of main effects and interactions.
The Introduction is too long and should focus on the use of SIT against the RPW.
The Summary and Abstract should focus on the results and their relevance to SIT.
The clarity of the Results section requires major improvement.
I have written suggestions and numbered points on a scanned copy of the manuscript.
NUMBERED POINTS (see scanned manuscript)
- 60% of the simple summary is preamble. Please focus on the results and their meaning
Authors: we agree with the reviewer, and we changed the simple summary.
- Ditto for the Abstract.
Authors: we agree with the reviewer and we changed the abstract.
- Two long paragraphs on the importance of RPW as a pest. Please reduce it to a few lines, we understand that it's an important pest.
Authors: Ok, we shortened this part (see lines 49-67).
- Please focus on why it is difficult to control by conventional means.
Authors: This part is discussed in the lines 68-79.
- The key to the SIT is the release of MASSIVE numbers of sterile males, so that the majority of matings of wild females occur with sterile males. This is even more important in females remate.
Authors: Ok we made the suggested change at line 84.
- I disagree. If females have "an equal chance" of receiving sterile and fertile sperm the SIT technique will be extremely inefficient! If females have a 10:1 chance or more of receiving sterile sperm, the technique will probably be more effective.
Authors: Of course, you are correct. But there are 2 aspects that should be taken under consideration in this paper:
- this work was addressed only to evaluate IF sterile males can be sexually competitive with the fertile males (in other words, an evaluation of the “quality” of the performance of the sterile males). If the sterile males are not sexually competitive, even if you release a ratio 20:1 sterile:fertile it will be a failure.
- in the case of RPW (but also for other gregarious insect pests) we are suggesting (only in small scale contexts) to use a combination of mass trapping and SIT: collecting large numbers of adults, eliminating (at least from the environment) the females and re-introducing wild-type sterile males they will find less fertile RPW (both males and females) (see the new paragraphs, lines 620-630).
7. Should be "Cobalt-60 rays"
Authors: Ok, we made the suggested correction
- Age of males when irradiated? Age when tested? It was unclear to me whether the insects were reared through several generations in the laboratory or not?
Authors: We replied to this question providing some changes in the text: we added a part (on the new version of the manuscript, see lines 146-150; 151-157; 175-182).
9a. Glazes? Do you mean paint? What type of paint?
Authors: Yes, for marking we used an odorless water-based natural paint of the brand Benecos. We added details in the text about the paint used.
9b. What time of day were observations performed? It seems from the results that mating declined over time, so this is important to clarify.
Authors: We agree with the reviewer that it is important to specify the time. Visual observations have always been done at the same time, from 9 a.m. to 1 p.m. We have added this specification in the text.
- fecundation? You mean insemination?
Authors: Ok, we made the suggested correction.
- Table 1 indicates the number of mating events, correct? Please state this explicitly. No need to use codes (0,1) that you employed in your statistical analysis.
Authors: Table 1 indicate the number of minutes, so we changed in “Total elapsed time in minutes during mating or during inactivity periods...” See lines 254-255.
- Percentage values can be given in parentheses beside the number of matings.
Authors: We changed the tables according your suggestions now (done).
- Why do you use asterisks in the text? Delete them.
Authors: we followed your suggestion (done)
- Please indicate degrees of freedom for Chi2 values throughout the manuscript.
Authors: we followed your suggestion (done)
- Why give asterisks and points for P-values? If you wish to highlight significant terms, present the p-values in bold (normal text for non-significant values).
Authors: we followed your suggestion (done)
- Tendency is not the correct term. You mean the effect was borderline significant (i.e. P = 0.1-0.05).
Authors: we followed your suggestion (done)
- What point are you making here? This text seems vague. Is this an issue that should be discussed in the Discussion?
Authors: we referred to the table 4b and now we added this information that should provide more clarity. The last part of the period was deleted since it does not contain so relevant information statistically speaking.
- Table 6 title is difficult to understand. Title says n of observations per minute, but Table says count in minutes (as if it were a duration). Not a count I think.
Authors: OK we made changes in the table title. See lines 378-382.
- Table 8. Frequency of matings in minutes??? I find it hard to believe that insects were mating 2 to 10 times per minute! Table text says minutes/day. Sounds more like a duration of mating to me, i.e. "time spent in copulation". Again, this needs to be made clear.
Authors: thank you, we have changed in the text according to your suggestion “time spent in copulation”. We also changed the table eliminating a row and adding “min” as unit of measure. We hope it is more clear now.
- Means of doses?? This sounds incorrect.
Authors: We deleted “of doses” now.
21.a. What is the correct y-axis label for Fig 5?
Authors: we changed n. with “time (min)”. We also replaced “mating frequency” with “Time spent in copulation” in the title.
21b. Table 9. Mating frequency in number of minutes per hour? This makes no sense.
Authors: we deleted frequency and changed in: “Glmer model (family=negative binomial): the response variable was analyzed as number of minutes per hour”. See lines 417-422.
- Figure 6. Has the same y-axis issue as Fig 5.
Authors: we changed n. with “time (min)”. We also replaced “mating frequency” with “Time spent in copulation” in the title.
- Higher than what?
Authors: we rewrote the entire paragraph see lines 437-447.
- This paragraph is confusing.
We apologize for this. Now we rewrote this paragraph and we hope that it was more clear now: see lines 437-447.
- No significant differences in what exactly?
Authors: We rewrote in: “No significant differences between the irradiated males and the fertile males for the mating event duration were observed (Table 10), although a borderline significance towards longer mating episodes was found in the case of the irradiated male at 60 Gy in comparison to the fertile male “ See lines 448-454.
- I did not understand the importance of this text. Is this something that needs to be moved to the Discussion?
Authors: we agree with you, not so important, we deleted this paragraph now.
- Surely a simple contingency table or Fisher's exact test would provide a clear analysis of this frequency data, rather than analyzing proportions (using the GLM model).
Authors: Since the data have also a random component (there are 4 observations per experimental unit corresponding to the 3rd, 6th, 9th and 12th day), we could not correctly estimate errors by a Fisher’s exact test since this could lead to biased p-values (probably smaller). So we preferred to use the same model.
- Please clarify the y-axis labels on Fig 8a and 8b.
Authors: done.
- Lower than what?
Authors: we changed in: “Conversely, regarding the elapsed time before the first mating…, a significantly shorter time was recorded on the irradiated male at 80 Gy in comparison to the fertile male.” See lines 506-509 on the new version of the manuscript.
- See my comment 5; SIT is based on the propensity of fertile females to mate with sterile males which greatly outnumber fertile males.
Authors: Ok, we changed the text as suggested by the reviewer (see lines 519-520 on the new version of the manuscript).
- What bottleneck?
Authors: We made changes in the text so that it was clearer. See lines 582-587.
- Drastic shifts in what?
Authors: We made changes in the text so that it was clearer. See lines 582-587.
- Unclear whether laboratory rearing was performed over several generations or not, although this text indicates insects were wild-caught (and used immediately?).
Authors: Adult weevils were captured in the field and kept in rearing until the start of the experiment. Therefore, rearing was not done for multiple generations, the same individuals collected in the field are used in the experiment. These points have been clarified in the text at lines 588-593 of the new version of the manuscript.
- If RPW males were reared over several generations in the laboratory they will have been selected to mate rapidly with females, as happens in mass-reared tephritid flies.
Authors: As explained in the previous answer, the adults used in the experiment were not bred for several generations.
- You do not seem to consider the insemination capacity of irradiated males. i.e. attempted matings vs. successful matings.
Authors: To distinguish attempted mating from successful mating, we counted and analyzed only mating with a duration of at least 30 seconds (see lines 245-247). In any case, the aim of the present study was to analyze and understand the behavior of irradiated males and compare it with that of fertile males. So, we did not consider fecundity, which has already been analyzed in other studies (see Musmeci et al. 2018).
- This is a long text that should focus tightly on the results of the study. Please rewrite and reduce to two or three sentences, as your findings are simple and do not require a complex text.
Authors: Ok, we reduced the conclusion to 3 paragraphs as suggested by the reviewer.
- The titles of the tables in the annex can be improved for clarity. I was confused by Table A4.
Authors: Okay we improved the clarity of the table.
- Also confused by Table A6.
Authors: Okay we improved the clarity of the table.
Round 2
Reviewer 2 Report
Thank you for the response to my questions and comments. These revisions make the information easier to understand and more clearly describe how insects were handled prior to mating experiments. With the work now in better context, it is acceptable for publication.
Reviewer 3 Report
The authors have addressed my suggestions.
Requires some editing.